# Risk Factors Associated with the Development of Hospital-Acquired Infections in Hospitalized Patients with Severe COVID-19

**DOI:** 10.3390/antibiotics12071108

**Published:** 2023-06-27

**Authors:** Fernando Solís-Huerta, Bernardo Alfonso Martinez-Guerra, Carla Marina Roman-Montes, Karla Maria Tamez-Torres, Sandra Rajme-Lopez, Narciso Ortíz-Conchi, Norma Irene López-García, Guadalupe Yvonne Villalobos-Zapata, Andrea Rangel-Cordero, Janet Santiago-Cruz, Luis Fernando Xancal-Salvador, Steven Méndez-Ramos, Eric Ochoa-Hein, Arturo Galindo-Fraga, Alfredo Ponce-de-Leon, Maria Fernanda Gonzalez-Lara, Jose Sifuentes-Osornio

**Affiliations:** 1Instituto Nacional de Ciencias Médicas y Nutrición Salvador Zubirán, Department of Medicine, Mexico City 14080, Mexico; fernando.solish@incmnsz.mx; 2Instituto Nacional de Ciencias Médicas y Nutrición Salvador Zubirán, Department of Infectious Diseases, Mexico City 14080, Mexico; carla.romanm@incmnsz.mx (C.M.R.-M.); karla.tamezt@incmnsz.mx (K.M.T.-T.); sandra.rajmel@incmnsz.mx (S.R.-L.); luis.ponceg@incmnsz.mx (A.P.-d.-L.); 3Clinical Microbiology Laboratory, Instituto Nacional de Ciencias Médicas y Nutrición Salvador Zubirán, Department of Infectious Diseases, Mexico City 14080, Mexico; narciso.ortizc@incmnsz.mx (N.O.-C.); irene.lopezg@incmnsz.mx (N.I.L.-G.); yvonne.villalobosz@incmnsz.mx (G.Y.V.-Z.); andrea.rangelc@incmnsz.mx (A.R.-C.); janet.santiagoc@incmnsz.mx (J.S.-C.); fernando.xancals@incmnsz.mx (L.F.X.-S.); steven.mendezr@incmnsz.mx (S.M.-R.); fernanda.gonzalezl@incmnsz.mx (M.F.G.-L.); 4Instituto Nacional de Ciencias Médicas y Nutrición Salvador Zubirán, Hospital Epidemiology Department, Mexico City 14080, Mexico; eric.ochoah@incmnsz.mx (E.O.-H.); arturo.galindof@incmnsz.mx (A.G.-F.); 5Instituto Nacional de Ciencias Médicas y Nutrición Salvador Zubirán, General Direction, Mexico City 14080, Mexico

**Keywords:** COVID-19, SARS-CoV-2, health care-associated infection, cross infection, hospital-acquired pneumonia

## Abstract

Recognition of risk factors for hospital-acquired infections (HAI) in patients with COVID-19 is warranted. We aimed to describe factors associated with the development of HAI in patients with severe COVID-19. We conducted a retrospective cohort study including all adult patients admitted with severe COVID-19 between March 2020 and November 2020. The primary outcome was HAI development. Bivariate and multiple logistic regression models were constructed. Among 1540 patients, HAI occurred in 221 (14%). A total of 299 episodes of HAI were registered. The most common HAI were hospital-acquired/ventilation-associated pneumonia (173 episodes) and primary bloodstream infection (66 episodes). Death occurred in 387 (35%) patients and was more frequent in patients with HAI (38% vs. 23%, *p* < 0.01). Early mechanical ventilation (aOR 18.78, 95% CI 12.56–28.07), chronic kidney disease (aOR 3.41, 95% CI 1.4–8.27), use of corticosteroids (aOR 2.95, 95% CI 1.92–4.53) and tocilizumab (aOR 2.68, 95% CI 1.38–5.22), age ≥ 60 years (aOR 1.91, 95% CI 1.27–2.88), male sex (aOR 1.52, 95% CI 1.03–2.24), and obesity (aOR 1.49, 95% CI 1.03–2.15) were associated with HAI. In patients with severe COVID-19, mechanical ventilation within the first 24 h upon admission, chronic kidney disease, use of corticosteroids, use of tocilizumab, age ≥ 60 years, male sex, and obesity were associated with a higher risk of HAI.

## 1. Introduction

Increased susceptibility to secondary bacterial and fungal infections has been described after respiratory viral infections [1]. Severe COVID-19 is associated with immune dysfunction characterized by persistent cytokine release, inflammatory activation, and immunosuppression [2]. Endothelial dysfunction, vascular leakage, impaired mucus clearance, biofilm formation, and microbiome changes, along with COVID-19-associated immune dysfunction, are implicated in the increased susceptibility to secondary infections [1]. The prevalence of secondary infections among hospitalized patients with COVID-19 ranges from 4% to 25%, but it may reach 50% in non-survivors [3,4,5,6]. Bloodstream infections (BSI), hospital-acquired/ventilator-associated pneumonia (HAP/VAP) due to Gram-negative bacteria, and COVID-19-associated aspergillosis (CAPA) are the leading causes of hospital-acquired infections (HAI), occurring in 25%, 36%, and 10–35% of cases, respectively [4,7,8]. Antibiotic overuse in hospitalized patients with COVID-19 may have affected the ecological and epidemiological trends of HAI [9]. Additionally, increasing frequencies of multidrug-resistant (MDR) isolates have been reported [10].

Although immunomodulatory treatments for COVID-19, such as dexamethasone and tocilizumab, have been associated with lower mortality [11,12], they could also lead to an increased risk of secondary infections. The use of methylprednisolone, hepatic failure, and diabetes have been associated with increased susceptibility to secondary infections [4]. Secondary bacterial infections lead to decreased survival rates and are a leading cause of death among severe COVID-19 patients [13]. Infections due to MDR pathogens have been associated with a mortality of 60% [14]. Recognition of risk factors for HAI in patients with COVID-19 is warranted. In this cohort study, we aimed to describe the risk factors associated with the development of HAI in hospitalized patients with severe and critical COVID-19.

## 2. Results

During the study period, 2017 patients were hospitalized with COVID-19, and 477 patients were excluded (Figure 1). We included 1540 patients who had a median age of 55 (IQR 45-65) years; 941/1540 (61.1%) were men. Obesity, hypertension, and type 2 diabetes mellitus (T2DM) were present in 681/1540 (44.2%), 528/1540 (34.3%), and 440/1540 (28.6%) subjects, respectively. The median time from symptom onset to admission was seven days (IQR 5-10). Empirical antibiotic therapy upon admission was used in 914/1540 (59.4%). Treatment with corticosteroids was prescribed in 688/1540 (44.6%) and tocilizumab in 97/1540 (6.3%). Among patients who developed HAI, male sex, obesity, lower oxygen saturation, lymphopenia, and higher concentrations of C-reactive protein (CRP), lactate dehydrogenase (LDH), and ferritin at hospital admission were present. Also, IMV and treatment with corticosteroids during the first 24 h of hospitalization were more frequent among patients with HAI. Enrolment in a COVID-19 clinical trial was more frequent in those without HAI (Table 1).

A total of 299 episodes of HAI occurred among 221 patients: 250/299 (83.6%) bacterial and 49/299 (16.3%) invasive fungal infections were registered. The most common HAI was HAP/VAP (173/299 episodes, 57.8%). The most frequently isolated organisms in respiratory samples were *Enterobacter cloacae* complex (49/173, 28.3%), *Pseudomonas aeruginosa* (35/173, 20.2%), and *Staphylococcus aureus* (32/173, 18.5%). A total of 66/299 (22.1%) episodes of primary bloodstream infections (BSI) were diagnosed; coagulase-negative *Staphylococcus* spp. (26/66, 39.4%) and *Enterococcus* spp. (12/66, 18.2%) were most frequently recovered. Additional bacterial infections included bone, joint, skin, and soft tissue infections (4/299 episodes, 1.3%), intraabdominal infections (3/299 episodes, 1.0%), urinary tract infections (2/299 episodes, 0.7%), and *Clostridioides difficile* infections (1/299 episode, 0.3%). A total of 26/299 episodes (8.7%) of probable/proven CAPA were documented, and *Aspergillus fumigatus* was the most common isolate (11/26, 42.3%). Additionally, 17/299 (5.7%) episodes of candidemia and 3/299 (1.0%) episodes of mucormycosis were diagnosed; *Candida parapsilosis* was the most common species isolated (12/17, 70.6%). A total of 19/221 (8.6%) and 9/221 (4.1%) patients were diagnosed with HAP/VAP and CAPA, HAP/VAP, and candidemia during their follow-up. Appendix A summarize bacterial and fungal pathogens recovered.

No carbapenem resistance was found among *Enterobacter* cloacae complex species (62 isolates), regardless of the site of infection. The second most frequent Gram-negative rod was *Escherichia coli* (46 isolates), and resistance to third-generation cephalosporins and carbapenems was observed in 23/46 (59.5%) and 4/46 (8.7%) isolates, respectively. Forty isolates of *P. aeruginosa* were recovered; piperacillin-tazobactam and carbapenem resistance was reported in 9/40 (22.5%) and 5/40 (12.5%) isolates, respectively. Among the Gram-positive cocci, the most common isolate was *S. aureus* (35 isolates); methicillin resistance was reported in 5/35 (14.3%) isolates. Among 23 *Candida* spp. isolates, 6/23 (26.1%) were azole-resistant (1 *C. glabrata* and 5 *C. parapsilosis*). Antimicrobial susceptibility patterns are summarized in Appendix A.

A total of 387/1540 (25.1%) patients died. Death occurred in 84/221 (38.8%) patients with HAIs and in 303/1319 (23.0%) without HAIs (*p* < 0.01). The presence of HAI was associated with a greater risk of death (RR 1.65, 95% CI 1.36–2.01). Treatment with corticosteroids was associated with a lower probability of death (RR 0.57, 95% CI 0.47–0.69). When stratified by HAI, the protective factor of corticosteroids for death was modified (Figure 2). HAP/HAP due to Gram-negative bacilli was associated with in-hospital death (RR 1.54, 95% CI 1.22–1.94). The latter was not observed when Gram-positive cocci caused HAP/VAP (RR 1.38, 95% CI 0.86–2.19). Both Gram-positive cocci and Gram-negative bacilli-associated BSI were not associated with in-hospital death (RR 1.40, 95% CI 0.93–2.10, and 1.33, 95% CI 0.72–2.45, respectively). CAPA was associated with in-hospital death (RR 1.96, 95%CI, 1.33–2.88). Candidemia was not associated with in-hospital death (RR 0.96, 95% CI 0.40–2.21).

Median LOS in survivors was seven days (IQR 5−14). Surviving patients that developed a HAI had longer LOS (median of 27 days [IQR 20−42] versus 7 days [5,6,7,8,9,10,11], *p* < 0.01). IMV during follow-up occurred in 131 patients, of which 58/131 (44.2%) developed HAI, and 73/131 (55.7%) did not (*p* = 0.187).

On bivariate analysis, a higher risk of HAI was associated with male sex, obesity, lower lymphocyte count, CRP > 10 mg/dL, ferritin > 500 mg/dL, LDH > 246 IU/L, D-dimer > 500 mg/mL, use of IMV on the first 24 h upon admission, and treatment with corticosteroids and tocilizumab. Enrolment in a COVID-19 clinical trial was associated with a lower risk of HAI (Appendix A). On multivariate analysis, use of IMV on the first 24 h upon admission (aOR 18.78, 95% CI 12.56–28.07), chronic kidney disease (aOR 3.41, 95% CI 1.4–8.27), treatment with corticosteroids (aOR 2.95, 95% CI 1.92–4.53) and treatment with tocilizumab (aOR 2.69, 95% CI 1.38–5.22), age greater than 60 years (aOR 1.91, 95% CI 1.27–2.88), male sex (aOR 1.52, 95% CI 1.03–2.24), and obesity (aOR 1.49, 95% CI 1.03–2.15) were independently associated with HAI (Table 2). Because of the strong association between the use of IMV in the first 24 h upon admission and the development of HAI, non-prespecified multivariate analysis using the previously selected variables was done to study the association between early IMV and HAP/VAP, BSI, candidemia, and CAPA. IMV in the first 24 h upon admission was independently associated with the development of HAP/VAP (aOR 15.86 95% CI 10.13–24.83), BSI (aOR 15.80 95% CI 7.94–31.42), CAPA (aOR 8.93 95% CI 3.56–22.40), and candidemia (aOR 7.27 95% CI 2.39–22.14) in separate models.

## 3. Discussion

In our study, IMV in the first 24 h, chronic kidney disease, treatment with corticosteroids, treatment with tocilizumab, older age, male sex, and obesity were independently associated with a higher risk of developing HAI in severe and critical COVID-19 hospitalized patients. Male sex has been associated with the development of secondary infections among COVID-19 patients in addition to a worse overall prognosis [15]. Up to 71% of patients that develop secondary infections are males [14], which may be related to sex-driven differences in immunological responses [16]. Similarly, age has been consistently associated with HAI and worse prognosis in COVID-19 [17]. The latter could be due to immunosenescence and increasing comorbidity burden, which have been described as risk factors for secondary infections in severe and critical COVID-19 [18]. Liver injury and T2DM were related to the development of BSI in an Italian cohort [4]. Similarly, T2DM, hypertension, and obesity were associated with an increased risk of secondary infection in COVID-19 patients [19]. Chronic kidney disease was associated with persistent immune dysfunction and a greater risk of HAI [20]. We did not observe an association between T2DM or hypertension and the development of HAI, which could be partly explained because of the high frequency of these comorbidities in our region [21] and, consequently, in our sample. Because of barrier disruption and ICU-related disease severity, IMV has been consistently described as one of the main risk factors for HAI [22]. In our study, early IMV may reflect increased severity or late presentation and, consequently, increased immune dysfunction on admission. Additionally, a longer time of airway exposure to biotrauma could have increased the risk of developing HAI. Treatment with corticosteroids and tocilizumab was associated with the development of HAI. The use of methylprednisolone has been previously associated with the development of secondary infections in COVID-19 patients [4]. Because of their immunomodulatory properties, corticosteroids and tocilizumab have been associated with the development of HAI [19]. Our results suggest that the beneficial effect of corticosteroid treatment on mortality could be mitigated by the presence of HAI. Secondary infections could disrupt the effects of corticosteroids and the immune response against SARS-CoV-2 [23].

The observed frequency of HAI is in accordance with previous reports [5,6,13,19]. An important difference must be established between co-infections, defined as those that occur in the first 72 h after hospital admission, and secondary infections, defined as those that occur after the first 48 h after hospital admission. Our focus was on secondary infections [3]. Higher mortality and LOS are expected in COVID-19 patients who develop HAI [14]. Our cohort is characterized by subjects with a high number of comorbidities, which are major problems in our region [21]. We also observed high proportions of empiric antibiotic therapy use, which has been previously described [24] and has not been associated with a lower risk of HAI [10]. Previous antibiotic exposure was not correlated with the antimicrobial susceptibility patterns of the infecting microorganisms.

HAP/VAP and primary BSI were the most frequently diagnosed HAIs. *Enterobacter* complex species, *E. coli*, *P. aeruginosa*, *S. aureus*, coagulase negative *Staphylococcus* species, and *Klebsiella pneumoniae* were the most commonly reported microorganisms, as previously described [10]. The main invasive fungal infection in our cohort was CAPA. The results of antimicrobial susceptibility testing are in accordance with previous COVID-19-related reports, where resistance to third-generation cephalosporins is a common problem and carbapenem resistance is an emerging threat [25]. Similarly, azole-resistant *Candida* infections are an emerging problem in hospital settings [26]. Antimicrobial overprescribing during the pandemic has increased dramatically. Inappropriate use of antimicrobials will likely play a role in the spread of antimicrobial resistance in the near future, yielding broad-spectrum antibiotics ineffective [27].

Our study presents limitations inherent to its retrospective nature that must be considered. We describe risk factors that predispose to a wide range of both bacterial and fungal infections, so the impact of individual risk factors may not reflect their association with a specific type of infection or individual microbial isolate. Even though our results suggest that early IMV is associated with HAI development, we cannot rule out that a longer time of exposure to IMV may have played a role in the development of infections. Of note, early IMV was independently associated with HAP/VAP, BSI, CAPA, and candidemia in exploratory analysis. Because our study was conducted before widespread vaccination was implemented in our region, the interpretation of our results must consider that a post-vaccination lower frequency of severe forms of COVID-19 could impact the development of HAIs. The importance of accurate imaging has been previously described [28], and although multilobe involvement was registered in most patients, we could not register additional radiological data. Additionally, the unicentric nature of the study must be taken into consideration. Finally, we cannot account for the learning curve during the study period, during which increasing knowledge and specific interventions (such as corticosteroid treatment) may have impacted our results. The large sample size, prespecified analysis plan, and systematic data collection are the main strengths of our study.

In this study, we describe risk factors associated with the development of various types of HAIs in patients hospitalized with severe forms of COVID-19. Although further research is warranted, recognizing such risk factors could allow active surveillance and continuous reinforcement of bundle prevention packages according to specific patient characteristics.

## 4. Materials and Methods

We conducted a retrospective cohort study in a COVID-19 reference center in Mexico City. Our center specializes in caring for adults with complex medical and/or surgical problems. On 16 March 2020, our center was converted into a dedicated COVID-19 facility. We registered demographic, clinical, and laboratory data of all consecutive adult patients admitted with a PCR-confirmed case of severe or critical COVID-19 between 18 March 2020 and 17 November 2020. We excluded patients who were transferred to other facilities before discharge, had a length of stay <24 h, or were discharged against medical advice. During the study period, routine SARS-CoV-2 real-time polymerase chain reaction (RT-PCR) testing on nasopharyngeal swab samples was performed upon admittance. Nucleic acid extraction was performed using the NucliSens easyMAG system (bioMérieux, Boxtel, The Netherlands), and RT-PCR was processed on an Applied Biosystems 7500 thermocycler (version 1.4, Foster City, CA, USA) according to specifications described elsewhere [29]. Data were registered using the electronic medical record. A severe case was considered when SpO_2_ was <93%, PaO_2_/FiO_2_ ratio < 300, respiratory rate ≥ 30 breaths per minute, or ≥50% lung involvement was observed in chest CT, whereas a critical case was considered when either shock, urgent need for invasive mechanical ventilation (IMV), or multi-organ failure were observed [30]. Immunosuppression was considered when the prescription of immunosuppressive medications or comorbidities (e.g., solid malignant tumors, hematologic malignancy, solid organ transplant, human immunodeficiency virus infection, and connective tissue disorders) were present. Each patient was followed up from admission to death or discharge. The primary outcome was the development of a culture-proven HAI. A HAI was considered if diagnosed 48 h after admission and if accepted criteria were met [31,32,33,34]. All cases were reviewed by an infectious disease specialist. During the study period, cultures and serum markers (e.g., procalcitonin and galactomannan) were only obtained if ordered by an infectious disease specialist. All samples were cultured for bacterial and fungal pathogens. Secondary outcomes were in-hospital death, length of stay (LOS), and IMV during follow-up. All patients received the standard of care according to available evidence at admission. In our center, the use of corticosteroids as standard treatment was implemented on 17 June 2020. During the study period, neither tocilizumab nor baricitinib was routinely administered. Written informed consent was waived due to the retrospective nature of the study. The study was approved by the Institutional Review Board (Ref. number 3333).

Considering a primary outcome probability of 50% [4], a mean absolute percentage error of 5%, and the identification of up to 10 potential predictors, we calculated a required sample size of at least 588 patients [35]. Mean, standard deviation (SD), median and interquartile range (IQR) were used to describe data according to the Shapiro–Wilk normality test. Comparisons between patients who presented the primary outcome were made using Chi-squared, Fisher’s exact test, independent samples t-test, and two-sample rank sum tests. To determine factors associated with the development of HAIs, bivariate analysis to calculate relative risk (RR) and 95% confidence interval (CI) were performed. A multiple logistic regression model that included variables of biological importance was constructed to identify factors independently associated with the development of HAI. Variables with interactions (e.g., liver cirrhosis, human immunodeficiency virus infection, and Charlson comorbidity index) assessed by the Mantel–Haenszel Chi test were not included. Laboratory values were not included in the multivariable analysis because of incomplete data and non-systematic determinations. For the multivariable analysis, adjusted odds ratios (aOR) were calculated. An exploratory analysis was done to describe the association between in-hospital death and the development of HAI. HAI-stratified associations between in-hospital death and treatment with corticosteroids were also studied in exploratory analysis. Missing data were not replaced. Two-sided *p* values < 0.05 were considered statistically significant. STATA version 15.1 (College Station, TX, USA) was used.

## 5. Conclusions

In our study, IMV within the first 24 h upon admission, chronic kidney disease, treatment with corticosteroids, treatment with tocilizumab, age greater than 60 years, male sex, and obesity were independently associated with a higher risk of presenting HAI in hospitalized patients with severe forms of COVID-19.

## Figures and Tables

**Figure 1 antibiotics-12-01108-f001:**
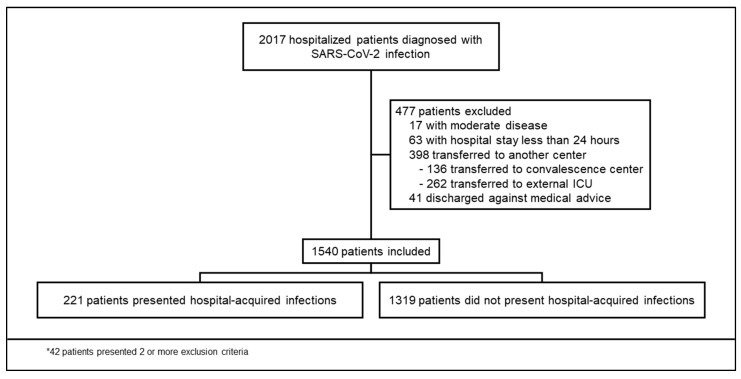
Enrolment and inclusion.

**Figure 2 antibiotics-12-01108-f002:**
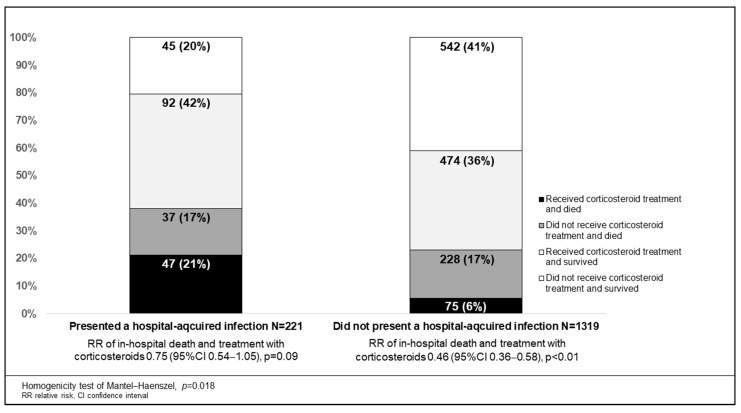
Association of in-hospital death and treatment with corticosteroids according to presence or absence of hospital-acquired infections.

**Table 1 antibiotics-12-01108-t001:** Baseline demographic and clinical characteristics.

	All Patients n = 1540 (100%)	Developed HAI n = 221 (14.35%)	Did Not Develop HAI n = 1319 (85.65%)	*p*
Male sex, n (%)	941 (61.1)	159 (72.0)	782 (59.3)	<0.001
Age, years—median (IQR)	55 (45–65)	56 (46–65)	54 (45–66)	0.633
Obesity—n (%) n = 1537	681 (44.3)	120 (54.3) n = 221	561 (42.3) n = 1316	0.001
Type 2 diabetes mellitus—n (%) n = 1539	440 (28.6)	68 (30.8) n = 221	372 (28.2) n = 1318	0.438
Hypertension—n (%) n = 1359	528 (34.3)	69 (31.2) n = 221	459 (34.8) n = 1318	0.296
Chronic obstructive pulmonary disease—n (%) n = 1539	22(1.4)	1 (0.5) n = 221	21 (1.6) n = 1318	0.351
Immunosuppression—n (%) n = 1538	87 (5.7)	13 (5.9) n = 220	74 (5.6) n = 1318	0.861
Cardiovascular disease—n (%) n = 1538	86 (5.6)	14 (6.4) n = 220	72 (5.5) n = 1318	0.590
Chronic kidney disease—n (%) n = 1539	51 (3.3)	8 (3.6) n = 221	43 (3.3) n = 1318	0.838
Liver cirrhosis—n (%) n = 1536	11 (0.7)	1 (0.5) n = 221	10 (0.8) n = 1315	1.000
Charlson comorbidity index >2—n (%)	482 (31.3)	70 (31.7)	412 (31.2)	0.897
Time from symptom onset to hospital admission, days—median (IQR)	7 (5–10)	7 (5–9)	7 (5–10)	0.974
Oxygen saturation <90%, n (%) n = 1510	1371 (90.8)	206 (97.2) n = 212	1165 (89.8) n = 1298	<0.001
Lymphocyte count <800 cells/uL%—n (%)	855 (55.9)	142 (64.8) n = 219	713 (54.4) n = 1311	0.004
C-reactive protein >10 mg/dL—n (%) n = 1498	1030 (68.8)	187 (86.6), n = 216	843 (65.8) n = 1282	<0.001
Ferritin >500 ng/mL—n (%) n = 1487	828 (55.7)	151 (72.3) n = 209	677 (53.0) n = 1278	<0.001
Lactate dehydrogenase ≥246 U/L—n (%) n = 1482	1285 (84.9)	198 (93.0) n = 213	1060 (83.5) n = 1269	<0.001
D-dimer >500 ng/mL—n (%) n = 1500	555 (37.0)	99 (46.1) n = 215	456 (35.5) n = 1285	0.003
Multilobe involvement in CT—n (%) n = 1538	1530 (99.5)	221 (100) n = 221	1309 (99.4) n = 1317	0.288
Use of invasive mechanical ventilation in the first 24 h—n (%)	279 (68.1)	143 (64.7)	136 (10.3)	<0.001
Empiric antibiotic therapy—n (%)	914 (59.4)	136 (61.5)	778 (59.0)	0.474
Treatment with steroids—n (%)	688 (44.7)	139 (62.9)	549 (41.6)	<0.001
Treatment with tocilizumab—n (%)	97 (6.3)	21 (9.5)	76 (5.8)	0.034
Enrollment in a COVID-19 clinical trial—n (%)	320 (20.8)	26 (11.8)	294 (22.3)	<0.001

dl deciliter, IQR interquartile range, L liters, mg milligrams, ml milliliter, ng nanograms, U units, uL microliter.

**Table 2 antibiotics-12-01108-t002:** Risk factors associated with the development of hospital-acquired infections.

	aOR (CI 95%) *p* *
Use of IMV in the first 24 h after admission	**18.78 (12.56–28.07)** ***p* < 0.0001**
Chronic kidney disease	**3.41 (1.40–8.27) *p* = 0.007**
Corticosteroid treatment	**2.95 (1.92–4.53) *p* < 0.0001**
Tocilizumab treatment	**2.69 (1.38–5.22) *p* = 0.004**
Age > 60 years	**1.91 (1.27–2.88) *p* = 0.002**
Male sex	**1.52 (1.03–2.24) *p* = 0.031**
Obesity	**1.49 (1.03–2.15) *p* = 0.031**
Immunosuppression	1.59 (0.75–3.35) *p* = 0.222
Cardiovascular disease	1.12 (0.52–2.39) *p* = 0.762
Diabetes mellitus	1.03 (0.68–1.54) *p* = 0.882
Empirical antibiotic therapy	1.00 (0.64–1.55) *p* = 0.993
Enrollment in a clinical trial	0.98 (0.59–1.64) *p* = 0.958
Oxygen saturation <90%	0.95 (0.33–2.69) *p* = 0.924
Hypertension	0.69 (0.46–1.05) *p* = 0.091
Chronic obstructive pulmonary disease	0.12 (0.01–1.05) *p* = 0.056
* 1429 observations, PseudoR^2^ = 0.2937.	

aOR adjusted odds ratio, CI confidence interval, IMV invasive mechanical ventilation.

## Data Availability

The data presented in this study are available upon reasonable request to the corresponding authors.

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
