# Peer review of "Risk Factors Associated with the Development of Hospital-Acquired Infections in Hospitalized Patients with Severe COVID-19"

_antibiotics, 2023, doi:10.3390/antibiotics12071108_

Round 1

Reviewer 1 Report

Good observational study on risk factors associated with development of hospital acquired infections in hospitalised patients with severe Covid 19 infection.

Relevant introduction and materials and Methods adequately describes inclusion of covid 19 patients and risk factors for hospital acquired Infection (HAI).

Results reflect real world evidence in Covid 19 institution.

Tables of results are comprehensive.

Discussion reflects on results and risk factors for HAI in Covid 19 patents. 

More comments,

  1. Factors related to Covid 19 hospital acquired infections was the predominant question addressed by research
  2.  Yes original research
  3.  Adds to the evidence in Covid 19 era
  4. Nil specific improvements required in observational analysis
  5. Conclusions are consistent with results
  6. references are appropriate
  7. Tables are conclusive

Appropriate quality of English

Author Response

Cover letter- Reply round 1

Bernardo A Martinez-Guerra, M.D., MSc.

Department of Infectious Diseases

Instituto Nacional de Ciencias Médicas y Nutrición Salvador Zubirán

15 Vasco de Quiroga, Belisario Domínguez Sección XVI, Tlalpan

 Mexico City, Mexico. 14080

Tel, +5255-5487-0900. Ext. 5869

beramg@gmail.com

bernardo.martinezg@incmnsz.mx

June 19, 2023

Dear Nattapong Thakham

Assigned Editor

Antibiotics

We hope this letter finds you well. We wish to resubmit an original article entitled “Risk factors associated with the development of hospital-acquired infections in hospitalized patients with severe COVID-19” for consideration to be published by Antibiotics. We gladly received the reviewers report. On behalf of the authors, I would like to thank you for your time and dedication. Also, on behalf of the authors, I would like to thank the reviewers for a thorough and helpful review. All the reviewers’ comments and suggestions are highly valuable to us. Their work has undoubtedly improved the quality of our article. Below, you will find a point-by-point response to each of the comments and suggestions.

We confirm that this work is original and has not been published, nor is it currently under consideration for publication elsewhere. All authors have reviewed and agreed with the contents of the manuscript. Additionally, we have no conflicts of interest to disclose.

Thank you for your consideration of this manuscript.

Kind regards,

Bernardo A Martinez-Guerra, M.D., MSc

Dear Reviewer 1,

I hope this reply finds you well. On behalf of the authors, I would like to thank you for your time and dedication. Your comments and suggestions are highly valuable to us. We are glad that our article was of your liking. We kindly thank you for your favourable review.

Kind regards,

Bernardo A Martinez-Guerra

Reviewer 2 Report

This retrospective study describes risk factors associated the development of various types of hospital-acquired infections (HAIs) in patients hospitalized with severe forms of COVID-19 in a center of Mexico.

The paper is well-structured and easy to follow. Congratulations on your research.

I wonder about the date on which the study was approved by the Institutional Review Board. In the Institutional Review Board Statement, you indicated that the Institutional Review Board of Instituto Nacional de Ciencias Médicas y Nutrición Salvador Zubirán approved the study on 26 March 2020. However, the study uses a retrospective design including patients from 18 March 2020 to 17 November 2020. It is surprising that the ethical approval took place at the beginning of the retrospective period. May you explain?

Some others comment for the authors to consider as below:

-        Line 139, you indicated 173 HAP/VAP, this result is not in accordance with Table S1 which indicates: "Hospital-acquired pneumonia/Ventilator-associated pneumonia 250 isolates in 176 episodes".

-        Lines 156-157: you indicated 156 episodes of S. aureus, this result is not in accordance with Table S3 which reported: "Staphylococcus aureus n=37".

-        For a better readability:

o   the risk factors should be sorted in descending order according to their aOR (lines 36-39 in the abstract, lines 183-188 in the results section, in Table 2, lines 197-198 in the discussion section and in the conclusion).

o   like “coagulase-negative Staphylococcus spp (26, 39.4%) and Enterococcus spp (12, 18.2%)”, add the percentages for all episodes reported lines 141 to 148 (BSI, intraabdominal infections, CAPA, candidemia…).

o   Line 151, regarding Enterobacter cloacae complex species, add the number of isolates (like performed regarding E coli in the next sentence). n=62 regarding Table S3.

-        According to the instructions of authors of Antibiotics, the materials and methods section must be moved after the discussion section.

-        Lines 93-94: the references must be cited using numbers.

Some typos:

-        Line 137: ":" instead of ";"

-        Line 159: Table “S3” instead of “S2”

-        Line 165: “HAP/VAP” instead of "HAI/HAP”

Author Response

Cover letter- Reply round 1

Bernardo A Martinez-Guerra, M.D., MSc.

Department of Infectious Diseases

Instituto Nacional de Ciencias Médicas y Nutrición Salvador Zubirán

15 Vasco de Quiroga, Belisario Domínguez Sección XVI, Tlalpan

 Mexico City, Mexico. 14080

Tel, +5255-5487-0900. Ext. 5869

beramg@gmail.com

bernardo.martinezg@incmnsz.mx

June 19, 2023

Dear Nattapong Thakham

Assigned Editor

Antibiotics

We hope this letter finds you well. We wish to resubmit an original article entitled “Risk factors associated with the development of hospital-acquired infections in hospitalized patients with severe COVID-19” for consideration to be published by Antibiotics. We gladly received the reviewers report. On behalf of the authors, I would like to thank you for your time and dedication. Also, on behalf of the authors, I would like to thank the reviewers for a thorough and helpful review. All the reviewers’ comments and suggestions are highly valuable to us. Their work has undoubtedly improved the quality of our article. Below, you will find a point-by-point response to each of the comments and suggestions.

We confirm that this work is original and has not been published, nor is it currently under consideration for publication elsewhere. All authors have reviewed and agreed with the contents of the manuscript. Additionally, we have no conflicts of interest to disclose.

Thank you for your consideration of this manuscript.

Kind regards,

Bernardo A Martinez-Guerra, M.D., MSc

Dear Reviewer 2,

I hope this reply finds you well. On behalf of the authors, I would like to thank you for your profound revision. All your comments and suggestions are highly valuable to us, as they have undoubtedly improved the quality of our work. Below, you will find a point-by-point response to your comments and suggestions.

Comment: I wonder about the date on which the study was approved by the Institutional Review Board. In the Institutional Review Board Statement, you indicated that the Institutional Review Board of Instituto Nacional de Ciencias Médicas y Nutrición Salvador Zubirán approved the study on 26 March 2020. However, the study uses a retrospective design including patients from 18 March 2020 to 17 November 2020. It is surprising that the ethical approval took place at the beginning of the retrospective period. May you explain?

Answer: In fact, the study was approved on 26 March 2020 as part of an institutional cohort study aimed to describe characteristics of COVID-19 admitted patients. The approval included the opportunity to check medical records in a retrospective fashion. Because of the latter, we did retrospectively included patients.  

Comment: Line 139, you indicated 173 HAP/VAP, this result is not in accordance with Table S1 which indicates: "Hospital-acquired pneumonia/Ventilator-associated pneumonia 250 isolates in 176 episodes".

Answer: the supplementary material has been corrected. As stated in the manuscript, the correct number of episodes is 173.

Comment: Lines 156-157: you indicated 35 episodes of S. aureus, this result is not in accordance with Table S3 which reported: "Staphylococcus aureus n=37".

Answer: the supplementary material has been corrected. As stated in the manuscript, the correct number of isolates is 35.

Comment: For a better readability: the risk factors should be sorted in descending order according to their aOR (lines 36-39 in the abstract, lines 183-188 in the results section, in Table 2, lines 197-198 in the discussion section and in the conclusion),   like “coagulase-negative Staphylococcus spp (26, 39.4%) and Enterococcus spp (12, 18.2%)”, add the percentages for all episodes reported lines 141 to 148 (BSI, intraabdominal infections, CAPA, candidemia…), Line 151, regarding Enterobacter cloacae complex species, add the number of isolates (like performed regarding E coli in the next sentence). n=62 regarding Table S3.

Answer: the risk factors in the abstract (lines 36-42), results section (lines 138-143), table 2, discussion section (lines 152-153) and conclusions section (lines 278-280) have been sorted accordingly. Percentages been added to episodes of BSI bone, joint, skin and soft tissue infections,  intraabdominal infections, urinary tract infections and Clostridioides difficile infections , CAPA, candidemia and mucormycosis have been added in (line 92, 96-100). The number of isolates of E. cloacae has been added in line 105-106. Denominators have been added in the manuscript to increase readability.

Comment:

According to the instructions of authors of Antibiotics, the materials and methods section must be moved after the discussion section.

Answer: the manuscript sections have been sorted accordingly.

Comment: Lines 93-94: the references must be cited using numbers.

Answer: the references have been corrected in line 247.

Comment: Some typos: Line 137: ":" instead of ";", Line 159: Table “S3” instead of “S2”, Line 165: “HAP/VAP” instead of "HAI/HAP”

Answer: the typos have been corrected.

Kind regards,

Bernardo A Martinez-Guerra

Reviewer 3 Report

Dear Authors

I would like to thank you for the opportunity of reviewing this interesting paper that is focused on a very remarkable and challenging topic that is a lively argument also in daily clinical practice. The COVID-19 pandemic has overwhelmed healthcare systems globally, imposing serious health, economic and social effects on the population. Bacterial and fungal co-infections are reported complications of COVID-19 in critically ill patients but may go unrecognized premortem due to diagnostic limitations. In the present study, the Authors described which factors are associated with the development of hospital-acquired infections in patients with severe COVID-19.

This paper is pleasurable to read, although it suffers from some limitations that Authors can easily adjust in order to improve their research making it more eligible for this important Journal. Furthermore, the Authors can improve some sections of the paper, adding information and including other important references about this topic that, in my opinion, should be cited and discussed. 

First of all, although the language used is appropriate, I (I am not a native English speaker) recommend to the Authors obtain a certified native speaker with proficiencies in the scientific-medical field to complete properly this paper (if not yet done). 

Although the introduction fits the context of the study, it is concise. Sometimes, many concepts clearly explicated in an exhaustive introduction could help readers to become passionate about reading the paper and using it as a reference. 

According to the literature, co-infections are defined as infections that occur ≤48–72 h after hospital admission and are rare in COVID-19 patients, being reported in about 7% of hospitalized patients. On the contrary, secondary infections (or superinfections) are defined as infections that emerge during the course of the illness or hospital stay (i.e., >48–72 h after admission) and are more frequently diagnosed, reaching up to 45% of cases. [doi:10.1016/j.jinf.2020.05.046][doi:10.1186/s13756-021-01024-4][doi:10.1016/j.chest.2021.04.002][doi: 10.1016/j.eimc.2021.10.014]. According to these definitions, could the Authors please specify if they evaluated co-infections or superinfections or both and state the appropriate definition in the text? Then, if necessary, please correct the title and the entire text, including the Materials and Methods section.

In the Materials and Methods section, the Authors state that: “The primary outcome was the development of a culture-proven HAI. A HAI was considered if accepted criteria were met”. Could the Authors please state in the text if blood samples underwent galactomannan test for the detection of aspergillosis infections and the criteria used for the diagnosis of pulmonary aspergillosis? [doi: 10.1016/S2213-2600(18)30274-1] [doi: 10.1164/rccm.201111-1978OC].

Since a concomitant fungal AND bacterial infection is the most common presentation in COVID-19 severely ill patients [doi:10.1016/j.chest.2021.04.002] [doi:10.1002/jmv.27548], could the Authors please report the number of patients with both fungal and bacterial infections in their cohort?

In the discussion, in my opinion, it is important to underline the role of imaging (in particular chest CT) in COVID-19 patients, since it represents one of the main techniques to assess the severity of the pneumonic infection. In fact, besides allowing prognosis estimation and helping in medical decision-making for hospitalization, imaging plays an important role in monitoring COVID-19 patients and detecting bacterial and fungal complications, thus suggesting further laboratory investigations. In particular, the presence of consolidations, cavitations, and bronchiectasis should be warning signs for radiologists, since they are associated with the presence of bacterial and/or fungal co-pathogens. Similarly, the presence or the de-novo appearance or the volumetric increase in bronchiectasis have been previously associated with fungal or mycobacterial colonization in COVID-19 patients [doi: 10.3390/diagnostics12040846]. Please discuss this topic and cite the aforementioned reference.

Finally, the Authors should further discuss the issue of antibiotic overprescribing in the COVID-19 population. Nosocomial infection-causing organisms have been increasingly reported to be resistant to common antibiotics and therapies, thus their successful management and proper treatment represent key factors to reduce hospitalization costs [doi: 10.1016/j.jinf.2020.05.046][doi: 10.1177/20499361221095732]. 

Kind regards,

Author Response

Cover letter- Reply round 1

Bernardo A Martinez-Guerra, M.D., MSc.

Department of Infectious Diseases

Instituto Nacional de Ciencias Médicas y Nutrición Salvador Zubirán

15 Vasco de Quiroga, Belisario Domínguez Sección XVI, Tlalpan

 Mexico City, Mexico. 14080

Tel, +5255-5487-0900. Ext. 5869

beramg@gmail.com

bernardo.martinezg@incmnsz.mx

June 19, 2023

Dear Nattapong Thakham

Assigned Editor

Antibiotics

We hope this letter finds you well. We wish to resubmit an original article entitled “Risk factors associated with the development of hospital-acquired infections in hospitalized patients with severe COVID-19” for consideration to be published by Antibiotics. We gladly received the reviewers report. On behalf of the authors, I would like to thank you for your time and dedication. Also, on behalf of the authors, I would like to thank the reviewers for a thorough and helpful review. All the reviewers’ comments and suggestions are highly valuable to us. Their work has undoubtedly improved the quality of our article. Below, you will find a point-by-point response to each of the comments and suggestions.

We confirm that this work is original and has not been published, nor is it currently under consideration for publication elsewhere. All authors have reviewed and agreed with the contents of the manuscript. Additionally, we have no conflicts of interest to disclose.

Thank you for your consideration of this manuscript.

Kind regards,

Bernardo A Martinez-Guerra, M.D., MSc

Dear Reviewer 3,

I hope this reply finds you well. On behalf of the authors, I would like to thank you for your thorough revision. All your comments and suggestions are highly valuable to us, as they have undoubtedly improved the quality of our work. We are especially thankful for providing further articles on specific topics related to our research. Below, you will find a point-by-point response to your comments and suggestions.

Comment: This paper is pleasurable to read, although it suffers from some limitations that Authors can easily adjust in order to improve their research making it more eligible for this important Journal. Furthermore, the Authors can improve some sections of the paper, adding information and including other important references about this topic that, in my opinion, should be cited and discussed. 

Answer: Thank you for your comments. Indeed, our study presents several limitations that we have stated in the discussion section. To increase readability, denominators have been added in the results section. References have been improved and the discussion expanded accordingly.  

Comment: First of all, although the language used is appropriate, I (I am not a native English speaker) recommend to the Authors obtain a certified native speaker with proficiencies in the scientific-medical field to complete properly this paper (if not yet done). 

Answer: a thorough revision has been undertaken and typos and grammar corrected.

Comment: Although the introduction fits the context of the study, it is concise. Sometimes, many concepts clearly explicated in an exhaustive introduction could help readers to become passionate about reading the paper and using it as a reference. According to the literature, co-infections are defined as infections that occur ≤48–72 h after hospital admission and are rare in COVID-19 patients, being reported in about 7% of hospitalized patients. On the contrary, secondary infections (or superinfections) are defined as infections that emerge during the course of the illness or hospital stay (i.e., >48–72 h after admission) and are more frequently diagnosed, reaching up to 45% of cases. [doi:10.1016/j.jinf.2020.05.046][doi:10.1186/s13756-021-01024-4] [doi:10.1016/j.chest.2021.04.002][doi: 10.1016/j.eimc.2021.10.014]. According to these definitions, could the Authors please specify if they evaluated co-infections or superinfections or both and state the appropriate definition in the text? Then, if necessary, please correct the title and the entire text, including the Materials and Methods section.

Answer: The references have been added in the introduction and further discussed in the discussion section in lines 180-184. The fact that only patients with secondary infections were studied has been clarified in the methods section in lines 246-249.

Comment: In the Materials and Methods section, the Authors state that: “The primary outcome was the development of a culture-proven HAI. A HAI was considered if accepted criteria were met”. Could the Authors please state in the text if blood samples underwent galactomannan test for the detection of aspergillosis infections and the criteria used for the diagnosis of pulmonary aspergillosis? [doi: 10.1016/S2213-2600(18)30274-1] [doi: 10.1164/rccm.201111-1978OC].

Answer: an updated definition of HAI has been provided in the manuscript. Routine processes in our centre have been clarified in lines 247-249.

Comment: Since a concomitant fungal AND bacterial infection is the most common presentation in COVID-19 severely ill patients [doi:10.1016/j.chest.2021.04.002] [doi:10.1002/jmv.27548], could the Authors please report the number of patients with both fungal and bacterial infections in their cohort?

Answer: the corresponding data has been included in the results sections in lines 101-103.

In the discussion, in my opinion, it is important to underline the role of imaging (in particular chest CT) in COVID-19 patients, since it represents one of the main techniques to assess the severity of the pneumonic infection. In fact, besides allowing prognosis estimation and helping in medical decision-making for hospitalization, imaging plays an important role in monitoring COVID-19 patients and detecting bacterial and fungal complications, thus suggesting further laboratory investigations. In particular, the presence of consolidations, cavitations, and bronchiectasis should be warning signs for radiologists, since they are associated with the presence of bacterial and/or fungal co-pathogens. Similarly, the presence or the de-novo appearance or the volumetric increase in bronchiectasis have been previously associated with fungal or mycobacterial colonization in COVID-19 patients [doi: 10.3390/diagnostics12040846]. Please discuss this topic and cite the aforementioned reference.

Answer: the fact that we did not register all radiological data constitutes a limitation of our study. We have discussed these matters in the discussion section lines 211-213.

Finally, the Authors should further discuss the issue of antibiotic overprescribing in the COVID-19 population. Nosocomial infection-causing organisms have been increasingly reported to be resistant to common antibiotics and therapies, thus their successful management and proper treatment represent key factors to reduce hospitalization costs [doi: 10.1016/j.jinf.2020.05.046][doi: 10.1177/20499361221095732]. 

Answer: this topic has been further discussed and referenced in lines 197-200.

Kind regards,

Bernardo A Martinez-Guerra

Round 2

Reviewer 3 Report

The Authors addressed raised points adequately.